# Left-digit bias in out-hospital cardiac arrest: The JCS-ReSS study

**Takahiro Suzuki[1], Atsushi Mizuno[1,2,3]***, Daisuke Yoneoka[4], Takahiro Nakashima[5], Tetsuya Matoba[6], Koichi Node[7], Naohiro Yonemoto[8], Yoshio Tahara[8], Yoshio Kobayashi[8], Takanori Ikeda[8]**

1 Department of Cardiovascular Medicine, St. Luke's International Hospital, Tokyo, Japan, 2 Tokyo Foundation for Policy Research, Tokyo, Japan, 3 Leonard Davis Institute for Health Economics, University of Pennsylvania, Philadelphia, Pennsylvania, United States of America, 4 Center for Surveillance, Immunization, and Epidemiologic Research, National Institute of Infectious Diseases, Tokyo, Japan, 5 Department of Emergency Medicine and The Max Harry Weil Institute for Critical Care Research and Innovation, University of Michigan, Ann Arbor, Michigan, United States of America, 6 Department of Cardiovascular Medicine, Kyushu University, Fukuoka, Japan, 7 Department of Cardiovascular Medicine, Saga University, Saga, Japan, 8 The Japanese Circulation Society with Resuscitation Science Study (JCS-ReSS) Group, Tokyo, Japan

* atmizu@gmail.com

**Data Availability Statement:** Data cannot be shared publicly, and only the results of the research will be shared. The dataset is provided to researchers through the JCS-ReSS, sent from the Fire and Disaster Management Agency of Japan's

## Abstract

### Introduction

The left-digit bias (LDB), a numerical-related cognitive bias, not only potentially influences decision-making among the general public but also that of medical practitioners. Few studies have investigated its role in out-of-hospital cardiac arrest (OHCA).

### Methods

We retrospectively included all consecutive patients with OHCA witnessed by family members registered in the All-Japan Utstein Registry of the Fire and Disaster Management Agency between January 1, 2005, and December 31, 2020. Target outcomes were the percentage of bystander cardiopulmonary resuscitation (BCPR) performed by family members or paramedics and the percentage of prehospital physician-staffed advanced cardiac life support (ACLS). Using a nonparametric regression discontinuity methodology, we examined whether a significant change occurred in the percentages of BCPR and ACLS at the age thresholds of 60, 70, 80, and 90 years, which would indicate the presence of LDB.

### Results

Of the 1,930,273 OHCA cases in the All-Japan Utstein Registry, 384,200 (19.9%) cases witnessed by family members were analyzed. The mean age was 75.8 years (±SD 13.7), with 38.0% (n = 146,137) female. We identified no discontinuities in the percentages of chest compressions, mouth-to-mouth ventilation, or automated external defibrillator (AED) usage by family members for the age thresholds of 60, 70, 80, and 90 years. Moreover, no discontinuities existed in the percentages of chest compressions, advanced airway management,

Ministry of Internal Affairs and Communications. While certain measures have been taken to anonymize the data, it cannot be guaranteed that individuals cannot be identified based on the location of fire stations, symptoms, etc. Therefore, access to this dataset and individual records is limited only to researchers. The Data Access Committee, Ethics Committee, or other institutions to which data requests are submitted is the secretariat of the Japanese Circulation Society (https://www.j-circ.or.jp/topics/jcs_notice_23112401/).

**Funding:** This research was partially supported by AMED under Grant Number JP 23rea522009h0001, awarded to AM. The funder had no role in study design, data collection and analysis, decision to publish, or preparation of the manuscript.

**Competing interests:** The authors have declared that no competing interests exist.

and AED usage by paramedics or prehospital ACLS by physicians for any of the age thresholds.

## Conclusions

In conclusion, our study did not find any evidence that age-related LDB affects medical decision-making in patients with OHCA.

## Introduction

Cognitive biases play a significant role in human decision-making, particularly affecting rational choices. Recently, cognitive biases have been reported to influence decision-making in the medical field, affecting not only patients but also healthcare professionals, posing a significant challenge to the provision of high-quality medical care [1]. Left-digit bias (LDB), one of the cognitive biases, is the tendency to classify continuous variables based on the left-most digit [2]. Despite the identical numerical difference, consumers perceive the difference between $4.00 and $2.99 as being larger than the difference between $4.01 and $3.00 [3]. Indeed, the presence of LDB has been reported in the medical field as well, and medical professionals tend to perceive the age difference between patients aged 80 and 79 as riskier than the difference between patients aged 78 and 79. Olenski et al. demonstrated that in the case of Coronary Artery Bypass Grafting following acute myocardial infarction, surgeons perceive a higher risk of surgical complications in patients aged 80 compared to those aged 79, potentially leading to more conservative treatment approaches for patients aged 80 [4]. Furthermore, previous studies have explored the impact of LDB on clinical decision-making regarding pre-surgical suitability for operations such as kidney transplants [5] and acute cholecystitis surgeries [6] across various clinical situations.

Generally, cognitive biases, including LDB, are considered to exist in scenarios where decision-making needs to be rapid. While the presence of LDB has consistently been demonstrated in pre-surgical suitability assessments, interestingly, a previous study examining the effects of LDB on outcomes in patients with in-hospital cardiac arrest (IHCA) found no evidence of LDB influencing survival or medical decision-making, such as the duration of resuscitation [7]. This suggests that the impact of cognitive biases, especially LDB, on medical professionals' decision-making may vary depending on the content and situation, particularly the time allowed for decision-making. It has been reported in other areas that the impact of LDB is affected by the response time allotted for decisions [3], suggesting that the influence of LDB might differ between scenarios where decisions are required within seconds, such as in resuscitation, and those where decisions are made over hours or days, such as in determining surgical suitability. Theoretically, in hyperacute situations, such as cardiac arrest, decision-making is often rapid and based on limited information, making these scenarios particularly susceptible to cognitive bias. However, the decision-making of healthcare providers and the public in out-of-hospital cardiac arrest (OHCA) situations has not been evaluated to date. OHCA is a critical issue significantly affecting patient outcomes, and it is crucial to clarify the impact of LDB on medical practices. Further, assessing these situations enables us to clarify the actual situation of LDB and the differences among decision-makers. Therefore, this study aims to verify the presence of LDB in OHCA situations by elucidating the influence of LDB on the frequency of medical practices performed by family members and healthcare professionals.

## Materials and methods

### Study setting

This cohort study used the nationwide, prospective, population-based OHCA registration system managed by the All-Japan Utstein Registry of the Fire and Disaster Management Agency (FMDA) [8]. Generally, in Japan, termination of resuscitation by paramedics before hospital arrival is not performed; thus, most OHCA patients treated by paramedics are transported to the hospital, and related data are recorded in the All Japan Utstein registry [9]. The details regarding the registration with the All Japan Utstein by the FDMA have been previously reported [8]. The data sheet is filled out based on information obtained from the patient and their family by the paramedics. Moreover, limited to cases of OHCA witnessed by family, it was possible to ascertain age in principle for all cases, and further, Japanese paramedics obtain basic information such as age and gender for all cases before arriving at the location of OHCA. Consequently, the cases where everyone performing resuscitation could more accurately grasp age were only those witnessed by family members. Therefore, we targeted this cohort for analysis. The Institutional Review Board of St. Luke's International Hospital approved this study (approval number: 22-R088).

### Study population and eligibility criteria

We retrospectively included all consecutive patients with OHCA witnessed by family members between January 1, 2005, and December 31, 2020. Fig 1 shows the study population of this study; we excluded 1,136,416 patients without witness, 404,394 patients witnessed by someone other than family members, 15 patients missing age data, and 5248 patients aged under 19 or over 105, resulting in a final cohort of 384,200 patients. We collected data on patients' age, sex, witnesses, and initial rhythms. The Utstein record is an administrative document that is normally maintained by the fire department, and does not contain information that can identify individuals. In other words, the document is anonymized, and it is not possible to obtain consent from each individual subject.

### Primary and secondary outcomes

The primary outcomes were the percentage of bystander cardiopulmonary resuscitation (CPR) (chest compressions, mouth-to-mouth ventilation, and automated external defibrillator (AED) usage) by family members, resuscitation by paramedics (starting chest compression by paramedics, advanced airway management (AAM) including laryngeal mask or

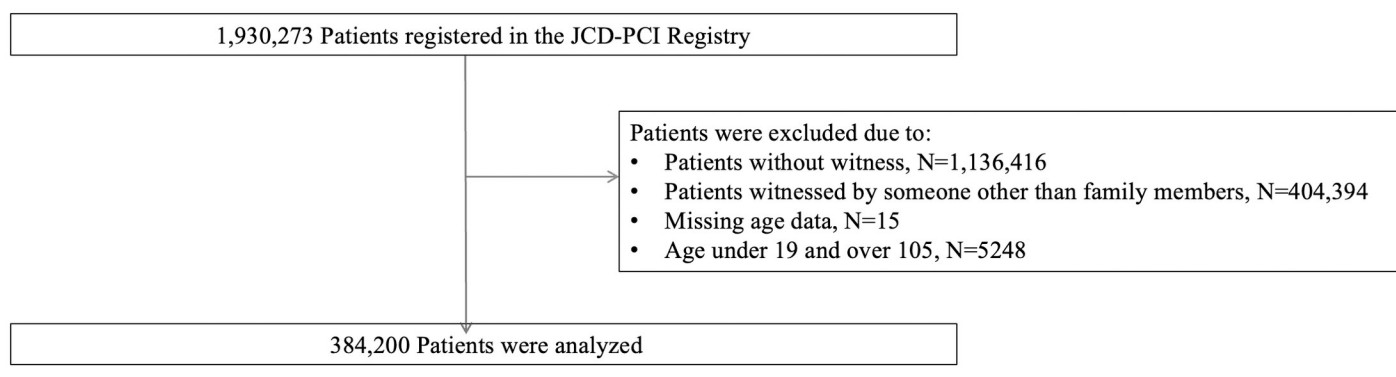

**Fig 1. Flow of the study population.**

esophagogastric tube airway, and AED usage), and prehospital advanced cardiac life support (ACLS) by physicians.

## Statistical analysis

Continuous variables were presented as mean ± standard deviations (SDs) or medians [interquartile ranges], and categorical variables are presented as proportions (counts). The comparing of baseline characteristics was performed by using the one-way ANOVA test for continuous variables and the Chi-squared test for categorical variables. LDB was assessed using a nonparametric regression discontinuity methodology. We tested whether the percentages of specific outcomes changed at the age thresholds of 60, 70, 80, and 90 years. The kernel width was estimated using error optimization. Furthermore, to demonstrate the robustness of our results, we conducted a sensitivity analysis comparing the frequency of resuscitation among patients aged 59 versus 60 years, 69 versus 70 years, 79 versus 80 years, and 89 versus 90 years, using both crude and adjusted models that account for age and initial rhythm. Additionally, to investigate the relationship between our findings and adjacent ages in more detail, we also conducted similar analyses for ages around the threshold of 80 years, such as comparing 78 versus 79 years and 80 versus 81 years. We extended this approach to similarly examine ages around the thresholds of 60, 70, and 90 years. In accordance with prior studies, this approach assumes that when comparing patients just below and just above an age threshold, it is assumed that both measured and unmeasured confounding factors are similar [7]. All analyses were conducted using R version 4.2.3 (R Foundation for Statistical Computing, Vienna, Austria), and a two-sided P-value < 0.05 threshold was used to determine statistical significance.

## Results

### Baseline information

Of 1,930,273 OHCA cases in the All-Japan Utstein Registry, 384,200 (19.9%) were analyzed according to the aforementioned exclusion criteria. Table 1 presents the clinical characteristics and initial arrest rhythm for each age group in 10-year increments. The mean age in the cohort was 75.8 years (±SD 13.7), with 38.0% (n = 146,137) female. Table 2 represents the frequency of resuscitation acts by family members, paramedics, and physicians for each age group in 10-year increments.

### Left-digit bias

Regarding family members' outcomes, we identified no discontinuities for bystander CPR at the age of 60, 70, 80, or 90 years for patients with OHCA (chest compression: coefficients,

**Table 1. Patient characteristics by age group in 10-year increments.**

| Characteristics | 20–29 years | 30–39 years | 40–49 years | 50–59 years | 60–69 years | 70–79 years | 80–89 years | 90- years |
|---|---|---|---|---|---|---|---|---|
| Number of patients | 2432 | 5802 | 13213 | 24540 | 55372 | 105231 | 128008 | 49602 |
| Gender (Female), n (%) | 879 (36.1) | 1867 (32.2) | 4372 (33.1) | 7659 (31.2) | 16963 (30.6) | 33051 (31.4) | 50980 (39.8) | 30366 (61.2) |
| Type of rhythm, n (%) | | | | | | | | |
| Ventricular fibrillation | 651 (26.8) | 1857 (32.0) | 3898 (29.5) | 6523 (26.6) | 11332 (20.5) | 12551 (11.9) | 8322 (6.5) | 1792 (3.6) |
| Pulseless ventricular tachycardia | 9 (0.4) | 19 (0.3) | 38 (0.3) | 83 (0.3) | 171 (0.3) | 325 (0.3) | 355 (0.3) | 123 (0.2) |
| Pulseless electrical activity | 505 (20.8) | 1197 (20.6) | 3163 (23.9) | 6760 (27.5) | 17865 (32.3) | 37187 (35.3) | 47491 (37.1) | 17565 (35.4) |
| Asystole | 1130 (46.5) | 2533 (43.7) | 5674 (42.9) | 10399 (42.4) | 24449 (44.2) | 52319 (49.7) | 68406 (53.4) | 28881 (58.2) |
| Others | 137 (5.6) | 196 (3.4) | 440 (3.3) | 775 (3.2) | 1555 (2.8) | 2849 (2.7) | 3434 (2.7) | 1241 (2.5) |

**Table 2. Frequency of each resuscitation act by age group in 10-year increments.**

| Overall frequency of CPR by citizens, n (%) | 20–29 years | 30–39 years | 40–49 years | 50–59 years | 60–69 years | 70–79 years | 80–89 years | 90- years |
|---|---|---|---|---|---|---|---|---|
| Chest Compression | 1242 (51.1) | 2844 (49.0) | 6158 (46.6) | 10914 (44.5) | 23439 (42.3) | 42379 (40.3) | 54268 (42.4) | 22082 (44.5) |
| Ventilation | 332 (13.7) | 731 (12.6) | 1289 (9.8) | 2432 (9.9) | 4336 (7.8) | 7032 (6.7) | 8077 (6.3) | 2999 (6.0) |
| AED usage | 17 (0.7) | 41 (0.7) | 87 (0.7) | 179 (0.7) | 333 (0.6) | 317 (0.3) | 201 (0.2) | 60 (0.1) |
| Overall frequency of chest compression started by paramedics, n (%) | 957 (42.9) | 2370 (44.9) | 5501 (46.5) | 10961 (49.6) | 25095 (51.1) | 48878 (53.0) | 56018 (50.3) | 20941 (48.2) |
| Overall frequency of AAM by paramedics, n (%) | | | | | | | | |
| Laryngeal Mask | 126 (6.5) | 324 (6.8) | 710 (6.8) | 1435 (7.1) | 2988 (6.6) | 5534 (6.5) | 5801 (5.7) | 2021 (5.3) |
| Esophageal Obturator | 686 (35.5) | 1905 (40.2) | 4564 (43.9) | 8425 (41.8) | 19185 (42.2) | 36508 (42.6) | 42088 (41.4) | 14320 (37.8) |
| Others | 128 (6.6) | 361 (7.6) | 902 (8.7) | 1759 (8.7) | 4502 (9.9) | 9683 (11.3) | 12689 (12.5) | 4703 (12.4) |
| None | 994 (51.4) | 2151 (45.4) | 4219 (40.6) | 8535 (42.3) | 18735 (41.3) | 33921 (39.6) | 41124 (40.4) | 16819 (44.4) |
| Overall frequency of AED usage by paramedics, n (%) | 775 (33.3) | 2255 (40.5) | 4711 (37.5) | 7989 (34.0) | 14351 (27.4) | 17486 (17.7) | 13250 (11.2) | 3303 (7.2) |
| Overall frequency of prehospital ACLS usage by physicians, n (%) | 287 (11.8) | 773 (13.3) | 1644 (12.4) | 3143 (12.8) | 6601 (11.9) | 12153 (11.6) | 13904 (10.9) | 4811 (9.7) |

Abbreviations: NA, not available; CPR, cardiopulmonary resuscitation; AED, automated external defibrillator; ACLS, advanced cardiac life support; AAM, advanced airway management

0.003 [95% CI, -0.012 to 0.017], 0.004 [95% CI, -0.013 to 0.026], 0.001 [95%CI, -0.018 to 0.013], and 0.005 [95%CI, -0.000 to 0.015], mouth-to-mouth ventilation: coefficients, 0.001 [95% CI, -0.006 to 0.012], 0.001[95% CI, -0.013 to 0.012], -0.002[95%CI, -0.006 to 0.001], and -0.000 [95%CI, -0.005 to 0.005], AED usage: coefficients, 0.001 [95% CI, -0.002 to 0.004], -0.001 [95% CI, -0.003 to 0.001], -0.001 [95%CI, -0.002 to 0.000], and -0.000[95%CI, -0.001 to 0.001], respectively) (Fig 2).

Concerning paramedics' outcomes, we identified no discontinuities for ACLS by paramedics at age thresholds of 60, 70, 80, and 90 years (Chest compression started by paramedics: coefficients, -0.011 [95% CI, -0.035 to 0.012], -0.013 [95% CI, -0.048 to 0.013], 0.001 [95%CI, -0.016 to 0.026], and 0.003 [95%CI, -0.010 to 0.007], Advanced airway management: coefficients, -0.020 [95% CI, -0.050 to 0.003], -0.006 [95% CI, -0.021 to 0.003], -0.002 [95%CI, -0.013 to 0.010], and 0.000 [95%CI, -0.025 to 0.022], AED usage: coefficients, 0.003 [95% CI, -0.017 to 0.021], -0.002 [95% CI, -0.034 to 0.038], 0.000 [95%CI, -0.025 to 0.022], and 0.004 [95%CI, -0.014 to 0.019], respectively).

Regarding physician outcomes, we found no discontinuities in prehospital ACLS at the age thresholds of 60, 70, 80, and 90 years (coefficients, -0.005 [95% CI, -0.014 to 0.003], 0.001 [95% CI, -0.006 to 0.006], 0.001[95%CI, -0.011 to 0.015], and -0.002 [95%CI, -0.016 to 0.011], respectively). The detailed data regarding these results are also summarized in S1 Table.

In our sensitivity analysis, when directly comparing the rates of primary outcomes for ages 59 years vs. 60 years, 69 years vs. 70 years, 79 years vs. 80 years, and 89 years vs. 90 years, we found no significant differences between the groups (Table 3). This demonstrates the robustness of our study's results based on the regression discontinuity design. Additionally, further comparisons conducted for ages 58 years vs. 59 years, 60 years vs. 61 years, 68 years vs. 69 years, 70 years vs. 71 years, 78 years vs. 79 years, 80 years vs. 81 years, 88 years vs. 89 years, and 90 years vs. 91 years are presented in the S2 Table. Here, several variables showed statistically

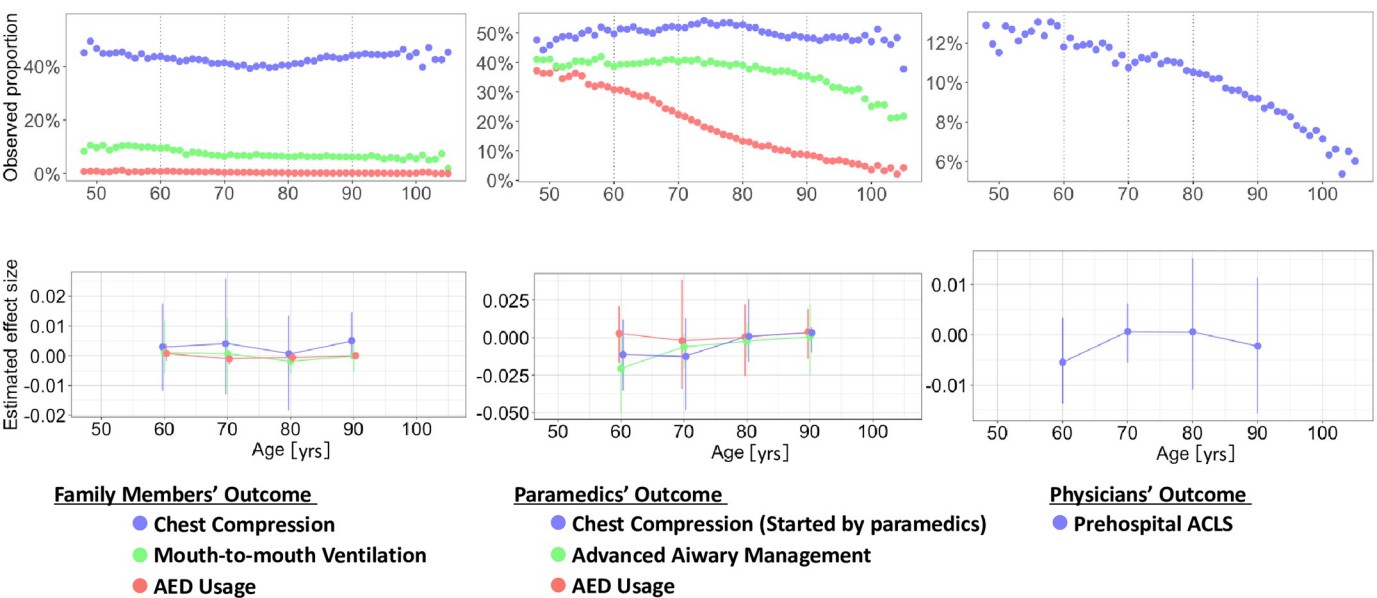

**Fig 2. Unadjusted probability of undergoing bystander-initiated cardiopulmonary resuscitation and prehospital physician-staffed advanced cardiac life support (ACLS).** In this model, we identified no discontinuities for bystander family CPR, paramedics' Chest compression, Advanced Airway management, AED usage, and physicians' ACLS at the age of 60, 70, 80, or 90 years. Abbreviations: NA, not available; CPR, cardiopulmonary resuscitation; AED, automated external defibrillator; ACLS, advanced cardiac life support.

significant differences between the two age groups in the frequency of resuscitator behaviors (S2 Table).

## Discussion

This is the first study to reveal an association between LDB and resuscitator behavior in patients with OHCA. This study provides the crucial insight that no demonstrated changes occurred in the practice of resuscitation at age thresholds of 60, 70, 80, and 90 years associated with LDB in a large nationwide cohort of OHCA patients. It is well-known that LDB does not consistently affect patient outcomes but contributes to physicians' decision-making [7, 10]. Age-related LDB in the medical field has been reported to affect determining factors such as eligibility for CABG [4], management of acute cholecystitis [6], selection of donors for kidney transplantation [5], and suitability for prostate cancer resection [11]. Conversely, LDB did not affect patient outcomes in cases of IHCA [7]. The strength of the Ututain Registry lies in the system in which all OHCA cases are transported to hospitals, as resuscitation termination decisions by paramedics are not feasible in Japan. To ascertain the impact of LDB on the behavior of resuscitators, we focused on the frequency of behaviors by family members, paramedics, and physicians rather than the clinical outcomes of patients. This enabled a more faithful evaluation of the verification of LDB in this field.

This study suggested that the age-related LDB does not affect the rate of resuscitation behaviors for OHCA, yet age is considered one of the important factors that could contribute to the implementation of resuscitation behavior and subsequent survival rates in cases of OHCA [12, 13]. The guidelines of the European Resuscitation Council indicate a lack of direct evidence regarding the impact of age on the outcomes following cardiopulmonary resuscitation [14], but past surveys have demonstrated that clinicians, regardless of their specialty, are less inclined to attempt cardiopulmonary resuscitation on patients aged 90 compared to those

**Table 3. Relative risk of probability of primary outcomes across left-digit thresholds.**

| Unadjusted | Threshold Age, years | Risk Ratio | 95%CI | p-value |
|---|---|---|---|---|
| Chest compression (Family members) | 60 vs. 59 | 1.00 | 0.95–1.06 | 0.92 |
| | 70 vs. 69 | 1.01 | 0.97–1.04 | 0.78 |
| | 80 vs. 79 | 1.00 | 0.97–1.03 | 0.87 |
| | 90 vs. 89 | 1.02 | 0.99–1.05 | 0.28 |
| Mouth-to mouth ventilation (Family members) | 60 vs. 59 | 0.98 | 0.85–1.13 | 0.81 |
| | 70 vs. 69 | 0.93 | 0.83–1.05 | 0.27 |
| | 80 vs. 79 | 0.95 | 0.87–1.04 | 0.30 |
| | 90 vs. 89 | 0.99 | 0.89–1.11 | 0.92 |
| AED usage (Family members) | 60 vs. 59 | 0.89 | 0.52–1.52 | 0.66 |
| | 70 vs. 69 | 0.62 | 0.36–1.05 | 0.07 |
| | 80 vs. 79 | 0.75 | 0.45–1.25 | 0.27 |
| | 90 vs. 89 | 0.93 | 0.48–1.80 | 0.83 |
| Chest compression (Paramedics) | 60 vs. 59 | 0.98 | 0.93–1.02 | 0.33 |
| | 70 vs. 69 | 0.99 | 0.96–1.03 | 0.72 |
| | 80 vs. 79 | 1.01 | 0.98–1.03 | 0.65 |
| | 90 vs. 89 | 0.99 | 0.96–1.02 | 0.66 |
| Advanced airway management (Paramedics) | 60 vs. 59 | 0.98 | 0.92–1.04 | 0.42 |
| | 70 vs. 69 | 0.98 | 0.95–1.02 | 0.36 |
| | 80 vs. 79 | 0.99 | 0.96–1.02 | 0.42 |
| | 90 vs. 89 | 1.00 | 0.97–1.04 | 0.86 |
| AED usage (Paramedics) | 60 vs. 59 | 0.97 | 0.90–1.04 | 0.41 |
| | 70 vs. 69 | 0.94 | 0.89–1.01 | 0.07 |
| | 80 vs. 79 | 0.93 | 0.82–1.04 | 0.43 |
| | 90 vs. 89 | 0.97 | 0.89–1.07 | 0.57 |
| ACLS (Physician) | 60 vs. 59 | 1.01 | 0.99–1.03 | 0.37 |
| | 70 vs. 69 | 1.00 | 0.99–1.01 | 0.82 |
| | 80 vs. 79 | 1.00 | 0.99–1.00 | 0.28 |
| | 90 vs. 89 | 1.00 | 0.99–1.01 | 0.74 |
| **Adjusted for sex and initial rhythm** | Threshold Age, years | Risk Ratio | 95%CI | p-value |
| Chest compression (Family members) | 60 vs. 59 | 1.01 | 0.92–1.11 | 0.86 |
| | 70 vs. 69 | 1.01 | 0.95–1.08 | 0.71 |
| | 80 vs. 79 | 0.99 | 0.95–1.04 | 0.82 |
| | 90 vs. 89 | 1.03 | 0.97–1.09 | 0.33 |
| Mouth-to mouth ventilation (Family members) | 60 vs. 59 | 0.98 | 0.84–1.15 | 0.84 |
| | 70 vs. 69 | 0.93 | 0.82–1.06 | 0.28 |
| | 80 vs. 79 | 0.95 | 0.86–1.04 | 0.27 |
| | 90 vs. 89 | 0.99 | 0.88–1.11 | 0.87 |
| AED usage (Family members) | 60 vs. 59 | 0.88 | 0.51–1.5 | 0.64 |
| | 70 vs. 69 | 0.61 | 0.36–1.04 | 0.07 |
| | 80 vs. 79 | 0.76 | 0.46–1.26 | 0.28 |
| | 90 vs. 89 | 0.93 | 0.48–1.81 | 0.83 |
| Chest compression (Paramedics) | 60 vs. 59 | 0.95 | 0.86–1.05 | 0.30 |
| | 70 vs. 69 | 0.98 | 0.92–1.05 | 0.63 |
| | 80 vs. 79 | 1.01 | 0.96–1.07 | 0.61 |
| | 90 vs. 89 | 0.99 | 0.93–1.05 | 0.69 |

*(Continued)*

**Table 3.** (Continued)

| | | | | |
|---|---|---|---|---|
| Advanced airway management (Paramedics) | 60 vs. 59 | 0.96 | 0.88–1.06 | 0.43 |
| | 70 vs. 69 | 0.97 | 0.91–1.04 | 0.40 |
| | 80 vs. 79 | 0.98 | 0.94–1.03 | 0.47 |
| | 90 vs. 89 | 1.01 | 0.95–1.07 | 0.80 |
| AED usage (Paramedics) | 60 vs. 59 | 0.96 | 0.81–1.14 | 0.66 |
| | 70 vs. 69 | 0.98 | 0.87–1.10 | 0.73 |
| | 80 vs. 79 | 0.94 | 0.85–1.04 | 0.25 |
| | 90 vs. 89 | 1.01 | 0.86–1.15 | 0.87 |
| ACLS (Physician) | 60 vs. 59 | 1.02 | 0.93–1.11 | 0.75 |
| | 70 vs. 69 | 0.99 | 0.90–1.09 | 0.82 |
| | 80 vs. 79 | 1.04 | 0.97–1.13 | 0.27 |
| | 90 vs. 89 | 1.02 | 0.93–1.11 | 0.75 |

aged 60 [15]. Recent reports have shown that, in the case of in-hospital cardiac arrests, there is more than a 20% difference in survival rates post-arrest between patient groups under 60 and those over 90 years of age, and for out-of-hospital cardiac arrests, advanced age is said to be an independent predictor of mortality in OHCA patients over 70 [15]. From this, it can be considered that an increase in patient age for OHCA might decrease the rate of resuscitation efforts, and age is a significant factor affecting resuscitation behaviors. However, in our data, age-related LDB was not observed in the context of OHCA. This cohort was limited to cases with a family witness, and by using data from a prospective registry that inquires in detail about age, gender, and other factors at the time of resuscitation, it was possible to more sensitively detect the presence of LDB. Considering the robustness of the results together with the sensitivity analysis, it was possible to obtain results that support, similar to previous studies on IHCA, that there is no age-related LDB in the very acute phase of cardiac arrest.

Several factors may account for the lack of recognition of LDB in cases of cardiac arrest, despite the established presence of LDB in surgical indications and organ transplantation. First, the impact of LDB on decision-making may vary depending on the context and situation, and the effect may be influenced by decision-time [3]. Surgical procedures such as CABG, cholecystectomy, or organ transplantation are typically planned strategies and have a time window of hours to days, allowing physicians and patients themselves sufficient time to choose the appropriate treatment methods. However, resuscitation is time-sensitive, and in cases of IHCA, the response should be an immediate emergency. In such situations, the urgency and rapid deterioration of a patient's condition during cardiac arrest may diminish the influence of LDB. Second, decision-makers and available alternatives differ in surgical and resuscitation decision-making. In the preoperative setting, patients are provided with sufficient information by physicians and can make decisions regarding their health by considering the risks and benefits of the treatment options [16]. However, in emergencies, such as cardiac arrest, patients have little opportunity to be involved in decision-making, and healthcare providers or family members need to make decisions within a short period of time. In such situations, the speed of decision-making and sense of urgency take priority, potentially diminishing the presence of LDB. Furthermore, in cases of surgery, potential improvement exists through alternative options, such as conservative treatment. However, the non-performance or interruption of resuscitation measures directly affects the patient's life, leaving little room for LDB in the decision-making process. Third, when physicians consider surgical options, decision-making may be susceptible to LDB because of age-specific risk scores and guideline recommendations. For

example, the GRACE score, a risk stratification scale used to assess acute coronary syndrome, calculates the risk score by dividing the age into 10-year intervals [17]. However, in cases of OHCA where no definitive age-based cutoff guideline exists, potential absence of age-related LDB is possible. Algorithms for responding to cardiac arrest aim to provide consistent patient care, commonly known as the BLS and ACLS [18]. In other words, responses to cardiac arrest are protocol-driven and not influenced by the patient's age or other individual characteristics. This approach may reduce the influence of LDB. Future studies should explore the effect of decision-making timeframes on age-related LDB. Understanding how urgent situations affect the consideration of age in decision-making can shed light on the potential presence or absence of LDB in these critical scenarios.

## Limitations

This study has several limitations. First, in cases witnessed by family members, we assumed that they knew the exact age of the patients but did not consider the possibility that they did not. Additionally, we did not obtain information on whether family members were instructed to inform non-family members, such as paramedics and physicians, of the patient's age before making decisions regarding resuscitation actions. However, as previously mentioned, in Japan, a system is in place where age is in principle solicited in advance across all cases of OHCA witnessed by family members. Therefore, by focusing on cases witnessed by family members, we could more accurately assess the impact of LDB. Second, unmeasured confounding factors were not considered. However, this method assumes that patients below and above the threshold have similar unmeasured confounders. Third, birth date information was unavailable. This study evaluated the presence of LDB at one-year intervals. In a previous study on CABG, patients were compared before and after their birthdays, which demonstrated the existence of an "LDB" effect. Furthermore, a previous report showed that patients who spent their birthdays in the intensive care unit received more intensive life-prolonging care and had longer ICU stays, yet had no significant difference in mortality rates compared with controls [19]. If data can be compared within specific periods, such as two weeks around birth, it may be possible to detect LDB influenced by precise thresholds at 10-year intervals.

## Conclusions

In conclusion, our nationwide cohort study found no evidence that age-related LDB influences resuscitation procedures in patients with OHCA.

## Supporting information

**S1 Table. Unadjusted probability of resuscitator behavior by family members, paramedics, and physicians.**
(DOCX)

**S2 Table. Relative risk of probability of primary outcomes in other numeric pairs near the target age threshold.**
(DOCX)

## Acknowledgments

We appreciate all the EMS personnel and participating physicians in Japan and the FDMA for their generous cooperation in establishing and maintaining the database. We also appreciate

all members of the Japanese Circulation Society Resuscitation Science Study (JCS-ReSS)
Group.

## Author Contributions

**Conceptualization:** Takahiro Suzuki, Daisuke Yoneoka, Tetsuya Matoba, Koichi Node.

**Data curation:** Naohiro Yonemoto.

**Formal analysis:** Takahiro Suzuki, Atsushi Mizuno, Daisuke Yoneoka.

**Investigation:** Takahiro Suzuki, Atsushi Mizuno, Daisuke Yoneoka, Takahiro Nakashima.

**Methodology:** Takahiro Suzuki, Atsushi Mizuno, Daisuke Yoneoka, Takahiro Nakashima,
Tetsuya Matoba, Koichi Node.

**Project administration:** Takahiro Suzuki, Atsushi Mizuno, Daisuke Yoneoka, Takahiro Naka-
shima, Tetsuya Matoba, Koichi Node, Naohiro Yonemoto, Yoshio Tahara, Yoshio Kobaya-
shi, Takanori Ikeda.

**Supervision:** Atsushi Mizuno, Daisuke Yoneoka, Takahiro Nakashima, Tetsuya Matoba, Koi-
chi Node, Naohiro Yonemoto, Yoshio Tahara, Yoshio Kobayashi, Takanori Ikeda.

**Validation:** Daisuke Yoneoka.

**Visualization:** Takahiro Suzuki, Atsushi Mizuno.

**Writing – original draft:** Takahiro Suzuki.

**Writing – review & editing:** Atsushi Mizuno, Daisuke Yoneoka, Takahiro Nakashima, Tet-
suya Matoba, Koichi Node, Naohiro Yonemoto, Yoshio Tahara, Yoshio Kobayashi, Taka-
nori Ikeda.

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
