## [Decision Letter · Decision Letter 0]

6 Feb 2024

PONE-D-23-31816Left-digit bias in out-hospital cardiac arrest : The JCS-ReSS studyPLOS ONE

Dear Dr. Mizuno,

Thank you for submitting your manuscript to PLOS ONE. After careful consideration, we feel that it has merit but does not fully meet PLOS ONE’s publication criteria as it currently stands. Therefore, we invite you to submit a revised version of the manuscript that addresses the points raised during the review process.

We look forward to receiving your revised manuscript.

Kind regards,

Alex Jones Flores Cassenote, Ph.D.

Academic Editor

PLOS ONE

Journal Requirements:

3. One of the noted authors is a group or consortium "the Japanese Circulation Society Resuscitation Science Study (JCS-ReSS) Group". In addition to naming the author group, please list the individual authors and affiliations within this group in the acknowledgments section of your manuscript. Please also indicate clearly a lead author for this group along with a contact email address.

Additional Editor Comments:

This is a structured approach to assess cognitive bias (left digit bias) among caregivers who performed cardiopulmonary resuscitation on patients outside the hospital.

I would like to see a slightly more expanded introduction, with greater exposition of the rationale that led to the development of the tested hypothesis.

I would like to see a more structured methodological process. I understand that this sample comes from a large study, however I believe that it would be more useful in understanding the results to have a broader view of the population base and its characteristics.

I believe it would be useful to present the results of the models developed in tables. Reading the graph is limited only to the dynamics of the phenomenon.

In Table 1 there is a statistical procedure applied, however it is not possible to understand the statistical origin of the p values by reading the methods.

The implications of this study's findings are unclear to me, even after some reading.

Furthermore, I would like to draw special attention to two points that are very strong in the reviewers' assessment.

1) It would be valuable to conduct a sensitivity analysis in which the authors compare the adjusted likelihood of resuscitation for patients age 79 vs 80 (as well as 78 vs 79 and 80 vs 81). Given that this is a negative study, confirming the null finding with such an alternative analytic approach would strengthen the veracity of the study.

2) According to the authors, a case where the age factor could potentially play out a role in decision making was defined to be a case witnessed by family members (but the life-support assistance could be provided by family members, paramedics, or physicians). However, the authors did not provide adequate information for whether the relevant protocol did instruct (or lead to a high likelihood) that the patient’s age be made known (in this case) to non-family members like paramedics and physicians *before* such an emergency medical intervention decision (and whether the relevant protocol would even instruct that age be considered as an agent for decision making in this case). Thus, whether the current case-selection criterion is a valid definition remains unclear and needs to be addressed by the authors.

In addition, to the important contribution made by the reviewers, I believe that the points mentioned above and copied by me here are relevant for a final opinion. I hope that the authors can create a new version considering all the points addressed by the referees and this academic editor. I will be proud to review this important material again.

Reviewers' comments:

Reviewer's Responses to Questions

**Comments to the Author**

1. Is the manuscript technically sound, and do the data support the conclusions?

Reviewer #1: Yes

Reviewer #2: No

2. Has the statistical analysis been performed appropriately and rigorously? 

Reviewer #1: Yes

Reviewer #2: No

3. Have the authors made all data underlying the findings in their manuscript fully available?

Reviewer #1: Yes

Reviewer #2: No

4. Is the manuscript presented in an intelligible fashion and written in standard English?

Reviewer #1: Yes

Reviewer #2: Yes

5. Review Comments to the Author

Reviewer #1: In this retrospective study using Japanese registry data, the authors investigate whether there is evidence of a left-digit bias in patient age on the performance of bystander CPR and pre-hospital ACLS. Analyzing over 384,000 witnessed cardiac arrests, they found that there was no left-digit effect at thresholds of 60, 70, 80, or 90 years of age. The analytic approach is valid and the conclusions are balanced and appropriate. Overall, I agree that there is value to publishing a “negative” study in this space that helps elucidate the types of scenarios in which cognitive biases like left-digit bias are unlikely to play a role.

I have 2 minor comments and one potential sensitivity analysis that I would suggest addressing in the discussion:

-First, as show in Fig 1, the author found that there is no association between age and bystander CPR at all. The lack of discontinuities therefore must be better reconciled with this finding, in that the absence of a left-digit bias for this finding may also be due to the fact that age is not a salient variable in this decision making. To give an extreme/absurd corollary: we would never expect there to be a left-digit bias between fingernail length and bystander CPR because clearly no bystander would even consider fingernail length as a salient characteristic when determining whether CPR should be performed.

-Second, I think there needs to be much greater discussion about the fact that another plausible mechanism is that exact age may not be known to the bystander, thereby making it impossible for a left digit effect to play a role. This is mentioned in the limitations but should be expanded in the main portion of the discussion that covers potential mechanisms.

-It would be valuable to conduct a sensitivity analysis in which the authors compare the adjusted likelihood of resuscitation for patients age 79 vs 80 (as well as 78 vs 79 and 80 vs 81). Given that this is a negative study, confirming the null finding with such an alternative analytic approach would strengthen the veracity of the study.

Reviewer #2: This study investigated whether the left-digit bias (LDB) would affect resuscitation decisions made on individuals OHCA cases where ages of the patients might be a known parameter. Specifically, OHCA cases witnessed by family members from 2005 to 2020 in a case registry database were obtained and a nonparametric regression discontinuity design was used to assess whether “cut-off” values for age thresholds (i.e., 60, 70, 80, and 90) explained drops in the frequency of life-support actions taken on these OHCA cases by family members, paramedics, and physicians. Among all the outcome areas assessed, the authors reported to identify no discontinuity. Although this study has a few merits (e.g., being the first to investigate the effect of LDB in life-support decisions on OHCA cases), I would like to recommend that the authors make a major revision where a few matters be reconsidered as listed below:

The most important question I have for the authors has to do with their analytical method choice and approach, given the underlying concept for LDB. What really detects whether there is an influence by LDB here in this context is that numeral pairs with identical (or almost indistinguishable) magnitude differences but different leftmost digit pairs (e.g., a 69-vs.-70 age numeral pair vs. a 68-69 or 70-71 age numeral pair) are treated significantly different in the outcome domains (e.g., in the context of the present study, it would be a *more* significant drop between a 69-vs.-70 pair than the difference (if any) observed in the other two pairs). The current analytical approaches, by just trying to detect whether there is a discontinuity at cut-off values like 60, 70, etc., could not adequately address the central question regarding an LDB effect. I would suggest that the authors consider a different statistical method that compares the differences in the outcomes among these age value-pairs with the same magnitude differences but different leftmost digit pairs.

In addition, statistical significance was not reported. I would also invite the authors to double check the accuracy for all the reported coefficient values (e.g., on the last line on p.8 for one of the coefficients for family members’ performance of mouth-to-mouth ventilation: it was reported to be -0.002, while the 95% CI was bounded by two positive values).

A second question I have is whether the current domain area (i.e., emergency decisions of life support for individuals with cardiac arrests) is a theoretically motivated or meaningful one to test the effect of LDB on medical decisions. Given the nature of this kind of medical situations (and as the authors have also tackled), age-related factors may not even be taken into consideration. Thus, the hypothesis for why LDB as a cognitive bias could even potentially (or theoretically) influence the relevant decision outcomes in this domain area does not seem to be well grounded (or at least, this was addressed in the Introduction). Therefore, it is not clear why the authors chose this particular domain area for investigating the LDB effect. I would invite the authors to provide more information in the Introduction section to address this matter.

My last major question is about the case-selection criteria. According to the authors, a case where the age factor could potentially play out a role in decision making was defined to be a case witnessed by family members (but the life-support assistance could be provided by family members, paramedics, or physicians). However, the authors did not provide adequate information for whether the relevant protocol did instruct (or lead to a high likelihood) that the patient’s age be made known (in this case) to non-family members like paramedics and physicians *before* such an emergency medical intervention decision (and whether the relevant protocol would even instruct that age be considered as an agent for decision making in this case). Thus, whether the current case-selection criterion is a valid definition remains unclear and needs to be addressed by the authors.

6. PLOS authors have the option to publish the peer review history of their article (what does this mean?). If published, this will include your full peer review and any attached files.

Reviewer #1: No

Reviewer #2: No

---

## [Author Response · Author response to Decision Letter 0]

26 Mar 2024

Mar 20, 2024

Dr. Emily Chenette

Editor-in-Chief

PLoS ONE

Re: Manuscript ID: PONE-D-23-31816

Title: “Left-digit bias in out-hospital cardiac arrest: The JCS-ReSS study”

We thank Dr. Emily Chenette and anonymous reviewers for their helpful suggestions and insights. We largely agree with the points raised and considered all of them in the revised version of our manuscript. We look forward to working with you and the reviewers to move this manuscript closer to publication in PLoS ONE.

The manuscript has been rechecked and the necessary changes have been made in accordance with the reviewers’ suggestions. The responses to all comments have been prepared and attached herewith/given below. 

Editor Comments

Response: 

Thank you for your kind guidance. 

I have reformatted the new manuscript after reviewing the style guidelines of PLOS ONE.

Response:

This paper is subject to restricted data access, and only the results of the research will be shared. The dataset is provided to researchers through the JCS-ReSS, sent from the Fire and Disaster Management Agency of Japan's Ministry of Internal Affairs and Communications. While certain measures have been taken to anonymize the data, it cannot be guaranteed that individuals cannot be identified based on the location of fire stations, symptoms, etc. Therefore, access to this dataset and individual records is limited only to researchers.

Regarding ethical restrictions associated with the research, it has been approved by the ethics review board of St. Luke's International Hospital. As for legal restrictions, the research is conducted in compliance with the Personal Information Protection Law and other relevant legislation.

3. One of the noted authors is a group or consortium "the Japanese Circulation Society Resuscitation Science Study (JCS-ReSS) Group". In addition to naming the author group, please list the individual authors and affiliations within this group in the acknowledgments section of your manuscript. Please also indicate clearly a lead author for this group along with a contact email address.

Response:

The lead author of this study is “Takahiro Suzuki”, who is listed as the first author. 

Due to the large number of contributors from our group (JCS-ReSS), it is difficult to list everyone's names. Therefore, we have decided not to list the group in our authors’ list. Instead, we will only change the names of individuals belonging to the group, along with their affiliations and group name.

Additionally, one of the members of the JCS-ReSS, Yoshio Kobayashi, has left the group now. Would it be possible to remove this co-author from our authors’ list? If it proves difficult, it is alright to leave it as is. I would be grateful if you could inform me about this matter.

I would like to see a slightly more expanded introduction, with greater exposition of the rationale that led to the development of the tested hypothesis.

Response:

Thank you for your important comments. Indeed, the Introduction was too brief and it was difficult to understand the objective of our hypothesis, so I have made major revisions. 

Line: 75-107

Please refer to the〈Introduction Session〉.

Cognitive biases play a significant role in human decision-making, particularly affecting rational choices. Recently, cognitive biases have been reported to influence decision-making in the medical field, affecting not only patients but also healthcare professionals, posing a significant challenge to the provision of high-quality medical care[1]. Left-digit bias (LDB), one of the cognitive biases, is the tendency to classify continuous variables based on the left-most digit[2]. Despite the identical numerical difference, consumers perceive the difference between $4.00 and $2.99 as being larger than the difference between $4.01 and $3.00[3]. Indeed, the presence of LDB has been reported in the medical field as well, and medical professionals tend to perceive the age difference between patients aged 80 and 79 as riskier than the difference between patients aged 78 and 79. Olenski et al. demonstrated that in the case of Coronary Artery Bypass Grafting following acute myocardial infarction, surgeons perceive a higher risk of surgical complications in patients aged 80 compared to those aged 79, potentially leading to more conservative treatment approaches for patients aged 80[4]. Furthermore, previous studies have explored the impact of LDB on clinical decision-making regarding pre-surgical suitability for operations such as kidney transplants[5] and acute cholecystitis surgeries[6] across various clinical situations.

Generally, cognitive biases, including LDB, are considered to exist in scenarios where decision-making needs to be rapid. While the presence of LDB has consistently been demonstrated in pre-surgical suitability assessments, interestingly, a previous study examining the effects of LDB on outcomes in patients with in-hospital cardiac arrest (IHCA) found no evidence of LDB influencing survival or medical decision-making, such as the duration of resuscitation[7]. This suggests that the impact of cognitive biases, especially LDB, on medical professionals' decision-making may vary depending on the content and situation, particularly the time allowed for decision-making. It has been reported in other areas that the impact of LDB is affected by the response time allotted for decisions [3], suggesting that the influence of LDB might differ between scenarios where decisions are required within seconds, such as in resuscitation, and those where decisions are made over hours or days, such as in determining surgical suitability. Theoretically, in hyperacute situations, such as cardiac arrest, decision-making is often rapid and based on limited information, making these scenarios particularly susceptible to cognitive bias. However, the decision-making of healthcare providers and the public in out-of-hospital cardiac arrest (OHCA) situations has not been evaluated to date. OHCA is a critical issue significantly affecting patient outcomes, and it is crucial to clarify the impact of LDB on medical practices. Further, assessing these situations enables us to clarify the actual situation of LDB and the differences among decision-makers. Therefore, this study aims to verify the presence of LDB in OHCA situations by elucidating the influence of LDB on the frequency of medical practices performed by family members and healthcare professionals.

I would like to see a more structured methodological process. I understand that this sample comes from a large study, however I believe that it would be more useful in understanding the results to have a broader view of the population base and its characteristics.

Response:

Thank you for your constructive feedback. As you pointed out, we have changed the methodology to a structured methodological process and provided more detailed descriptions.

Please refer to the〈Methods Session〉.

Line: 109-158

Materials and methods 

Study Setting

This cohort study used the nationwide, prospective, population-based OHCA registration system managed by the All-Japan Utstein Registry of the Fire and Disaster Management Agency (FMDA) [8]. Generally, in Japan, termination of resuscitation by paramedics before hospital arrival is not performed; thus, most OHCA patients treated by paramedics are transported to the hospital, and related data are recorded in the All Japan Utstein registry [9]. The details regarding the registration with the All Japan Utstein by the FDMA have been previously reported [8]. The data sheet is filled out based on information obtained from the patient and their family by the paramedics. Moreover, limited to cases of OHCA witnessed by family, it was possible to ascertain age in principle for all cases, and further, Japanese paramedics obtain basic information such as age and gender for all cases before arriving at the location of OHCA. Consequently, the cases where everyone performing resuscitation could more accurately grasp age were only those witnessed by family members. Therefore, we targeted this cohort for analysis.

Study Population and Eligibility Criteria 

We retrospectively included all consecutive patients with OHCA witnessed by family members between January 1, 2005, and December 31, 2020. Figure 1 shows the study population of this study; we excluded 1,136,416 patients without witness, 404,394 patients witnessed by someone other than family members, 15 patients missing age data, and 5248 patients aged under 19 or over 105, resulting in a final cohort of 384,200 patients. We collected data on patients’ age, sex, witnesses, and initial rhythms. The Utstein record is an administrative document that is normally maintained by the fire department, and does not contain information that can identify individuals. In other words, the document is anonymized, and it is not possible to obtain consent from each individual subject.

Primary and Secondary Outcomes 

The primary outcomes were the percentage of bystander cardiopulmonary resuscitation (CPR) (chest compressions, mouth-to-mouth ventilation, and automated external defibrillator (AED) usage) by family members, resuscitation by paramedics (starting chest compression by paramedics, advanced airway management (AAM) including laryngeal mask or esophagogastric tube airway, and AED usage), and prehospital advanced cardiac life support (ACLS) by physicians.

Statistical Analysis

 Continuous variables were presented as mean ± standard deviations (SDs) or medians [interquartile ranges], and categorical variables are presented as proportions (counts). The comparing of baseline characteristics was performed by using the one-way ANOVA test for continuous variables and the Chi-squared test for categorical variables. LDB was assessed using a nonparametric regression discontinuity methodology. We tested whether the percentages of specific outcomes changed at the age thresholds of 60, 70, 80, and 90 years. The kernel width was estimated using error optimization. Furthermore, to demonstrate the robustness of our results, we conducted a sensitivity analysis comparing the frequency of resuscitation among patients aged 59 versus 60 years, 69 versus 70 years, 79 versus 80 years, and 89 versus 90 years, using both crude and adjusted models that account for age and initial rhythm. Additionally, to investigate the relationship between our findings and adjacent ages in more detail, we also conducted similar analyses for ages around the threshold of 80 years, such as comparing 78 versus 79 years and 80 versus 81 years. We extended this approach to similarly examine ages around the thresholds of 60, 70, and 90 years. In accordance with prior studies, this approach assumes that when comparing patients just below and just above an age threshold, it is assumed that both measured and unmeasured confounding factors are similar[7]. All analyses were conducted using R version 4.2.3 (R Foundation for Statistical Computing, Vienna, Austria), and a two-sided P-value < 0.05 threshold was used to determine statistical significance.

I believe it would be useful to present the results of the models developed in tables. Reading the graph is limited only to the dynamics of the phenomenon. In Table 1 there is a statistical procedure applied, however it is not possible to understand the statistical origin of the p values by reading the methods. The implications of this study's findings are unclear to me, even after some reading.

Response:

Thank you for your meaningful feedback. The current figures alone made it difficult to understand specific values and detailed data, so we have created Supplemental Table 1 as an addition. It contains detailed results of our regression discontinuity design. Furthermore, since the statistical methods in Table 1 only account for baseline differences among age groups, we omitted the mention of p-values to make differences in patient backgrounds across age groups clearer. For improved readability, the original Table 1 has been divided into Table 1 and Table 2 in the revised manuscript.

Please refer to the Table 1 and Supplemental Table1

Supplemental Table 1. Unadjusted probability of resuscitator behavior by family members, paramedics, and physicians

Variable Cutoff Coefficient 95%CI p-value

Chest compression (Family members) 60 0.003 -0.012 – 0.017 0.70

 70 0.004 -0.013 – 0.026 0.51

 80 0.001 -0.018 – 0.013 0.76

・・・ ・・・ ・・・ ・・・ ・・・

Furthermore, I would like to draw special attention to two points that are very strong in the reviewers' assessment. 1) It would be valuable to conduct a sensitivity analysis in which the authors compare the adjusted likelihood of resuscitation for patients age 79 vs 80 (as well as 78 vs 79 and 80 vs 81). Given that this is a negative study, confirming the null finding with such an alternative analytic approach would strengthen the veracity of the study. 2) According to the authors, a case where the age factor could potentially play out a role in decision making was defined to be a case witnessed by family members (but the life-support assistance could be provided by family members, paramedics, or physicians). However, the authors did not provide adequate information for whether the relevant protocol did instruct (or lead to a high likelihood) that the patient’s age be made known (in this case) to non-family members like paramedics and physicians *before* such an emergency medical intervention decision (and whether the relevant protocol would even instruct that age be considered as an agent for decision making in this case). Thus, whether the current case-selection criterion is a valid definition remains unclear and needs to be addressed by the authors.

Response:

For the two important comments you pointed out, we will respond to each comment in the Reviewer Comments section.

Please refer to the response#3 for Reviewer and the response #2 for Reviewer 2.

In addition, to the important contribution made by the reviewers, I believe that the points mentioned above and copied by me here are relevant for a final opinion. I hope that the authors can create a new version considering all the points addressed by the referees and this academic editor. I will be proud to review this important material again.

Response:

We sincerely appreciate the very positive feedback on our manuscript. Based on your advice, we have further improved the manuscript by adding analyses and making revisions.

Reviewer 1 Comments

Reviewer #1: In this retrospective study using Japanese registry data, the authors investigate whether there is evidence of a left-digit bias in patient age on the performance of bystander CPR and pre-hospital ACLS. Analyzing over 384,000 witnessed cardiac arrests, they found that there was no left-digit effect at thresholds of 60, 70, 80, or 90 years of age. The analytic approach is valid and the conclusions are balanced and appropriate. Overall, I agree that there is value to publishing a “negative” study in this space that helps elucidate the types of scenarios in which cognitive biases like left-digit bias are unlikely to play a role.

Response: 

Thank you very much for the very positive comments on this manuscript. Based on the advice received, we have made further improvements to the manuscript by adding analyses and making revisions.

Response:

#1.

-First, as show in Fig 1, the author found that there is no association between age and bystander CPR at all. The lack of discontinuities therefore must be better reconciled with this finding,

---

## [Decision Letter · Decision Letter 1]

3 Jun 2024

Left-digit bias in out-hospital cardiac arrest : The JCS-ReSS study

PONE-D-23-31816R1

Dear Dr. Mizuno,

We’re pleased to inform you that your manuscript has been judged scientifically suitable for publication and will be formally accepted for publication once it meets all outstanding technical requirements.

Kind regards,

Chiara Lazzeri

Academic Editor

PLOS ONE

Additional Editor Comments (optional):

Reviewers' comments:

Reviewer's Responses to Questions

**Comments to the Author**

1. If the authors have adequately addressed your comments raised in a previous round of review and you feel that this manuscript is now acceptable for publication, you may indicate that here to bypass the “Comments to the Author” section, enter your conflict of interest statement in the “Confidential to Editor” section, and submit your "Accept" recommendation.

Reviewer #1: All comments have been addressed

Reviewer #2: All comments have been addressed

2. Is the manuscript technically sound, and do the data support the conclusions?

Reviewer #1: Yes

Reviewer #2: Yes

3. Has the statistical analysis been performed appropriately and rigorously? 

Reviewer #1: Yes

Reviewer #2: Yes

4. Have the authors made all data underlying the findings in their manuscript fully available?

Reviewer #1: No

Reviewer #2: No

5. Is the manuscript presented in an intelligible fashion and written in standard English?

Reviewer #1: Yes

Reviewer #2: Yes

6. Review Comments to the Author

Reviewer #1: The authors have adequately addressed my concerns. This manuscript has limitations related to its observational nature, but the discussion covers these

Reviewer #2: The revised manuscript and the author's response have addressed all the questions I have previously raised.

7. PLOS authors have the option to publish the peer review history of their article (what does this mean?). If published, this will include your full peer review and any attached files.

Reviewer #1: No

Reviewer #2: No

---

## [Editor Report · Acceptance letter]

18 Jun 2024

PONE-D-23-31816R1 

PLOS ONE

Dear Dr. Mizuno, 

I'm pleased to inform you that your manuscript has been deemed suitable for publication in PLOS ONE. Congratulations! Your manuscript is now being handed over to our production team.

Kind regards, 

on behalf of

Dr. Chiara Lazzeri 

Academic Editor

PLOS ONE